# Calcium Superphosphate-Mediated Reshaping of Denitrifying Bacteria Community Contributed to N_2_O Mitigation in Pig Manure Windrow Composting

**DOI:** 10.3390/ijerph18010171

**Published:** 2020-12-29

**Authors:** Yaguo Jin, Yingcheng Miao, Yajun Geng, Mengyuan Huang, Yihe Zhang, Xiuchao Song, Shuqing Li, Jianwen Zou

**Affiliations:** 1Jiangsu Key Laboratory of Low Carbon Agriculture and GHGs Mitigation, College of Resources and Environmental Sciences, Nanjing Agricultural University, Nanjing 210095, China; 2015203017@njau.edu.cn (Y.J.); 2017103088@njau.edu.cn (Y.M.); 2017203042@njau.edu.cn (Y.G.); 2018103046@njau.edu.cn (M.H.); 2020203073@stu.njau.edu.cn (Y.Z.); jwzou21@njau.edu.cn (J.Z.); 2Institute of Agricultural Resources and Environment, Jiangsu Academy of Agricultural Sciences, Nanjing 210014, China; xiuchao103@163.com; 3Jiangsu Key Lab and Engineering Center for Solid Organic Waste Utilization, Jiangsu Collaborative Innovation Center for Solid Organic Waste Resource Utilization, Nanjing Agricultural University, Nanjing 210095, China

**Keywords:** calcium superphosphate (CaSSP), nitrous oxide (N_2_O), manure composting, denitrifiers, model

## Abstract

Composting is recognized as an effective strategy for the sustainable use of organic wastes, but also as an important emission source of nitrous oxide (N_2_O) contributing to global warming. The effects of calcium superphosphate (CaSSP) on N_2_O production during composting are reported to be controversial, and the intrinsic microbial mechanism remains unclear. Here, a pig manure windrow composting experiment lasting for ~60 days was performed to evaluate the effects of CaSSP amendment (5%, *w*/*w*) on N_2_O fluxes in situ, and to determine the denitrifiers’ response, and their driving factors. Results indicated that CaSSP amendment significantly reduced N_2_O emissions as compared to the control pile (maximum N_2_O emission rate reduced by 64.5% and total emission decreased by 49.8%). CaSSP amendment reduced the abundance of *nirK* gene encoding for nitrite reductase, while the abundance of *nosZ* gene (N_2_O reductase) was enriched. Finally, we built a schematic model and indicated that the abundance of *nirK* gene was likely to play a key role in mediating N_2_O production, which were correlated with NH_4_^+^-N and NO_3_^−^-N changing responsive to CaSSP. Our finding implicates that CaSSP application could be a potential strategy for N_2_O mitigation in manure windrow composting, and the revealed microbial mechanism is helpful for deepening the understanding of the interaction among N-cycle functional genes, physicochemical factors, and greenhouse gases (GHG) emissions.

## 1. Introduction

Composting is considered to be one of the effective ways for organic wastes disposal, which is presented as an environmentally friendly and cost-efficient process to manage and recycle organic solid wastes [1]. Nevertheless, gaseous nitrogen (N) losses constitute a significant challenge of manure composting, which are detrimental to the environment and can cause secondary pollution [2,3,4,5,6]). Nitrous oxide (N_2_O) is a potent greenhouse gas (GHG) that exhibits global warming potential of 298 times that of carbon dioxide (CO_2_) on a 100-year time horizon [7]. The N_2_O emissions from manure composting may constitute 0.2–9.9% of initial total nitrogen (TN) loss of raw materials [8,9].

Several strategies have been developed to mitigate gaseous emissions during composting, including aeration control [3,10,11,12], bulking agents [13], and amendment addition [14,15,16,17], as well as microbial inoculation [18]. Calcium superphosphate (CaSSP), a commonly used and low-cost phosphorus-containing additive fertilizer, has been proposed as a potential strategy to mitigate N loss from manure composting systems [19,20]. Luo et al. [15] and Yuan et al. [21] reported that the addition of CaSSP decreased N_2_O emissions by 22.2–55.4% during composting. By contrast, Yang et al. [22] showed that the CaSSP amendment increased N_2_O emissions during kitchen waste composting. These contradictory effects of CaSSP addition on N_2_O emissions may relate to the variation of physicochemical properties, and particular the involved microbial processes during composting.

It is well documented that the formation of N_2_O during composting were contributed to different microbial nitrogen-transforming pathways, including nitrification and denitrification processes under various oxygen conditions [23,24]. Nitrification is a two-step oxidation of ammonium (NH_4_^+^) to nitrate (NO_3_^−^) via nitrite (NO_2_^−^) by chemolithoautotrophic ammonia oxidizers and nitrite oxidizers, where ammonia monooxygenase encoding by *amoA* is known to an important enzyme in this process; while microbial denitrification represents the stepwise reduction of nitrate or nitrite to N_2_ covered by four denitrified genes (*narG*, *nirK*/*S*, *norB* and *nosZ*) [25,26,27]. It has been highlighted that nitrifying and denitrifying gene abundance could predict N_2_O emissions as potential proxies [28]. Importantly, these two nitrogen-transforming processes were regulated by various abiotic factors [29]. Changes in crucial environmental factors during composting can shape the N_2_O-related functional microbes [24], which will then directly affect N_2_O emissions. Several studies have documented that CaSSP addition during composting can affect the abiotic characteristics of composts, such as decreasing pH and shifting the NH_4_^+^-N and NO_3_^−^-N content, which was considered to be the primary factors driving the increase/decrease in N_2_O emissions [30,31]. However, the correlation between these changes in physicochemical factors and microbial community is ignored, and it is not yet clear how CaSSP addition can affect the behavior of microbial functional groups during composting. Therefore, more attention is needed to survey the response of functional microbial populations to CaSSP addition during composting, as well as the intermediary physicochemical factors, so as to understand the underlying mechanism that CaSSP addition affects N_2_O emissions.

In this study, commercial windrow composting was carried out to measure N_2_O fluxes in situ. The abundance of functional genes involved in N_2_O emissions were quantified by real-time quantitative PCR. The main objectives of this study were to (1) examine the dynamics of N_2_O emissions with CaSSP addition during composting; (2) identify the CaSSP-mediated key microbial pathways that affect N_2_O emissions during composting; (3) illustrate the CaSSP-responsive physicochemical factors that shape the functional microbial groups. We successfully identified that CaSSP addition mediated NH_4_^+^-N/NO_3_^−^-N transformation in pig manure composting, consequently suppressed the abundance of *nirK* genes, and significantly decreased N_2_O emissions, which firstly explains the microbiology mechanism involved in CaSSP-mediated N_2_O mitigation in composting.

## 2. Materials and Methods

### 2.1. Windrow Composting Experiment

The windrow composting system for ~60 days was conducted in a fertilizer company, located at Nanjing, Jiangsu province, China. Two treatments were carried out using pig manure mixed completely with vegetable (Broccoli) and wheat straw at a dry mass ratio of 6:1 as raw materials. In addition to the control treatment (CK), calcium superphosphate (CaSSP) was amended with a weight of 5% (*w*/*w*, dry weight (DW); according to Luo et al. [15] and Yuan et al. [21] of the raw composting materials in calcium superphosphate treatment (CaSSP treatment). Both treatments contained three windrow composts for replications. The volume of each composting pile was ~20 m^3^ (12 m in length, 1.5 m in width and 1 m in height). The composting process included two phases: phase I was the bio-oxidative phase with regularly turning by a composter turner every two days for ~30 days; phase II was the post-maturation phase without turning. Each pile was located for gas sampling respectively. Composting material samples were collected near the positions of gas sampling and included three replicates.

### 2.2. Gas Sampling and Flux Measurement

N_2_O fluxes were measured using closed static chamber system (diameter 0.45 m, height 5 m) [13,32]. For this, PVC chamber bases (30 cm length × 30 cm width × 25 cm height) were inserted 25 cm into the pile at 10–12 h before gas sampling. Gas samples were collected once a week. The opaque chamber was placed on the peak of each windrow with rim of the chamber fitted into the groove of collar. As sampling, the groove in the top edge of the collar was filled with water to seal the rim of the chamber (Appendix A). At 0, 5, 10, 20, and 30 min after the chamber closure, 60 mL of gas samples were extracted with an airtight syringe and immediately injected into 100 mL pre-evacuated gas sampling bags. Gas samples were taken mainly between 8:00 and 10:00 am on each sampling day and then transported to the laboratory for analysis by gas chromatograph (GC) within a few hours. Concentrations of N_2_O were measured by a Gas Chromatograph (Agilent 7890A, Agilent Technologies, Inc. Wilmington, USA) with a flame ionization detector (FID) and an electron capture detector (ECD) [32,33]. N_2_O fluxes were calculated by a linear regression as described by Hutchinson and Mosier [34]. Total N_2_O emissions were sequentially accumulated from the fluxes between every two adjacent measurements [32].

### 2.3. Physicochemical Properties Determination

During the aerobic composting process, windrow temperature at a depth of 30 cm was monitored by a mercury thermometer and recorded at each sampling day. Material samples were divided into three parts after passing through 2-mm sieve: one part was air-dried for total carbon (TC) and total nitrogen (TN) analyses. The other two parts were immediately stored at 4 °C and −80 °C for further analysis, respectively. Moisture content of fresh samples was determined as a weight loss upon drying at 105 °C in an oven for 24 h. Inorganic nitrogen, including ammonium (NH_4_^+^) and nitrate (NO_3_^−^), were extracted from material samples using 2 mol L^−1^ potassium chloride (KCl) solution and then analyzed using the segmented flow analyzer (Auto Analyzer 3, Seal, Germany). For pH measurement, the samples were diluted with deionized water at a volume ratio of 1:10 and analyzed by a pH electrode (PHS-3C mv/pH detector, Shanghai, China). The TC content, TN content and C/N ratio were determined using an element analyzer (Elementar Vario Macro cube, Hanau, Germany).

### 2.4. Real-Time Quantitative PCR (q-PCR) Assays of the Functional Genes

According to the manufacturer’s instructions, DNA samples were extracted from composting materials (days 1, 10, 24, 31, 45, and 60) in triplicate using the UltraClean soil DNA isolation kit (Qiagen, Hilden, Germany), and then stored at −20 °C for q-PCR assays. The Nanodrop (ThermoFisher Scientific, Waltham, MA, USA) was used to determine the quantity and quality of DNA samples. Functional genes associated with nitrification and denitrification were quantified from the composting process to elucidate the role of nitrifying and denitrifying bacteria on N_2_O dynamics. Ammonium monooxygenase gene (*amoA*) was selected as nitrify marker gene, which include the ammonia oxidizing bacteria (AOB) and archaea (AOA), while the nitric oxide reductase gene (*nirK*/*S*) and nitrous oxide reductase gene (*nosZ*) were targeted as denitrify marker genes. The standard curves were made using stepwise 10-fold diluted pMD-18 vectors with correct functional gene fragments. Each DNA sample was diluted with sterile water to 10 ng DNA µL^−1^ before q-PCR assays. Amplification was performed using StepOne Plus Real-time PCR System (Thermo Fisher Scientific, Waltham, MA, USA). PCR mixtures contained 10 µL SYBR@ Premix Ex Taq TM (Takara, Dalian, China), 0.4 µL forward and reverse primers, 0.4 µL ROX reference dye II (50×), 2 µL template DNA, and 6.8 µL sterile water with a final volume of 20 µL. Primers sequence and PCR reaction conditions were listed in Table 1.

### 2.5. Statistical Analysis

All data were reported as means and standard errors. After checking the normal distribution of results and the homogeneity of variances using Kolmogorov-Smirnov (K-S) and Shapiro-Wilk (S-W) test by SPSS 22.0 (SPSS Inc., Chicago, IL, USA), a Tukey post hoc test was used to determine the significant difference between Control and CaSSP treatments. Pearson correlation analyses between N_2_O fluxes, functional genes abundance and physicochemical properties were carried out by using SPSS 22.0. Structural equation modeling (SEM) analysis conducted using AMOS 22.0 (Amos, Development Corporation, Meadville, PA, USA) was established to address the direct and indirect effects of calcium superphosphate on the soil properties, functional genes abundance, and N_2_O emissions.

## 3. Results

### 3.1. N_2_O Fluxes

Over the 60-day period of manure composting, N_2_O emission patterns and temporal trends for the different treatments were similar (Figure 1a). At the beginning, N_2_O fluxes maintained a constantly low level, then increased sharply on day 52 and lasted until the end of the composting. However, the peak N_2_O emission rate from CaSSP treatment (58.32 mg m^−2^ h^−1^) was significantly lower than that from Control (164.46 mg m^−2^ h^−1^), resulting in a 64.5% reduction of the maximum N_2_O emission rate brought by CaSSP application. Consequently, CaSSP amendment significantly reduced total N_2_O emission over the total composting period by 49.8% (1.18 g m^−2^ vs. 2.35 g m^−2^; Appendix A).

### 3.2. Physical and Chemical Characteristics Measurement

Composting temperature in Control and CaSSP treatments generally showed similar variation patterns with a scope of 42 °C to 71 °C (Figure 1b), which went through three typical degradation phases (mesophilic, thermophilic and curing phases). For both treatments, the temperature increased and entered the thermophilic phase on day 10 and peaked on day 31 (68–71 °C). In addition, similar trends of moisture of the two treatments over the whole manure composting were observed. They decreased constantly to day 38 and then remained a stable level, with a range from 12–14% (Figure 1b).

The pH values of the two treatments remained at 6.8 to 7.6 during the whole composting process (Figure 2a). An increase in pH was noted in the control treatment within the first 10 days, and the peak was achieved at 7.5 on day 17. On the contrary, adding CaSSP decreased the pH due to its acidic feature.

The dynamic changes of TN with time were shown in Figure 2b. Along with the reduction of carbonaceous materials (Figure 2c), the TN content of the both treatments increased continuously. During composting, the rate of organic nitrogen mineralization was lower than that of organic carbon, causing the C/N ratio of the two treatments to decrease (Figure 2d).

As shown in Figure 3a,b, soil NO_3_^−^-N and NH_4_^+^-N contents ranged from 0.21 to 0.77 mg N kg^−1^ and 1.96 to 3.21 mg N kg^−1^, respectively, which showed a trade-off between each other during the composting, as NH_4_^+^-N contents generally revealed a decreased pattern along the time while that for NO_3_^−^-N contents was opposite. As expected, NH_4_^+^-N contents of CaSSP treatment were slightly higher than the Control, which could be attributed to that the lower pH caused by CaSSP addition hindered the conversion of NH_4_^+^-N to NH_3_. The NO_3_^−^-N concentration was distinctly lower than the NH_4_^+^-N concentration during the composting of CK and CaSSP. The NO_3_^−^-N concentration in CK and CaSSP piles decreased from 0.33 and 0.29 g kg^−1^ DW to 0.26 and 0.22 g kg^−1^ DW after the first 10 days composting, respectively (Figure 3b). The high temperature inhibited nitrifying bacteria activity and maintained the low NO_3_^−^-N concentration in the both treatments from day 10–30. Afterwards, the NO_3_^−^-N concentration in CK and CaSSP treatments slowly elevated until the end of the experiment.

### 3.3. Quantification of the Functional Genes Involved in N_2_O Emission

The nitrifying genes (AOA *amoA* and AOB *amoA*) and denitrifying genes (*narG*, *nirK*, *nirS* and *nosZ*) were quantified in the composting process (Figure 4). The copy numbers of AOB *amoA* in CK and CaSSP piles increased during the first 10 days composting, and then gradually decreased. After 31 days, a slightly increase was observed and the highest value occurred on day 60 of composting (Figure 4a); similar changing trend was observed in AOA *amoA* in both treatments (Figure 4b). The AOB *amoA* abundance in CaSSP treatment was higher than that in the Control pile, especially between 10 and 50 days; while the AOA *amoA* population was found to be similar in both treatments (Figure 4a,b). The abundance of *nirS* and *nirK* generally remained a consistently level at the beginning of the composting, and then showed a continuous increase and reached the peak finally (Figure 4c,d). Importantly, after 30 days, both the population of *nirS* and *nirK* in Control was observed to be significantly higher than that in CaSSP treatment (*p* < 0.05) (Figure 4c,d). On the contrary, the abundance of *nosZ* increased dramatically on day 10 and then showed a continuous decrease until the end of experiment; the Control pile revealed a lower *nosZ* population than CaSSP treatment before 30 days (Figure 4e).

### 3.4. Linking N_2_O Emissions to the Abundance of N-Cycling Functional Microbial Genes in CaSSP-Amended Compost

The N_2_O emissions were correlated with the abundance of N-cycling microbial functional genes and soil properties (Table 2). The N_2_O cumulative emissions measured in the CaSSP treatment displayed significant positive correlations with the abundance of *nirK* (r = 0.79, *p* < 0.05). However, correlation analyses showed that the N_2_O cumulative emissions shared negative correlations with pile moisture, Total C, C/N ratio, and NH_4_^+^-N.

To better understand the effect of CaSSP on N_2_O emissions linked to the nitrifier and denitrifier functional gene abundance, path analysis was used to examine the interactions among the variables (Figure 5). From this, the addition of CaSSP had direct and significant influence on NH_4_^+^-N, whereas indirectly influenced *nirK* abundance via changing NO_3_^−^-N. Therefore, the abundance of the *nirK* gene showed a significant and direct effect on N_2_O emissions (λ = 0.97, *p* < 0.05).

## 4. Discussion

The present study was performed to better understand the process by addressing the effects of CaSSP amendment on N_2_O mitigation during composting, as well as improve the mechanistic understanding of the role of microbial denitrification contributing to this mitigation.

N_2_O emissions in Control and CaSSP treatments observed in maturity stage during composting (Figure 1a), which may be attributed to accumulated NO_3_^−^-N during the thermophilic phase. It has been reported that high temperature (above 40 °C) and high concentrations of free ammonia may inhibit nitrifiers [12,40]. Here, it was observed that the CaSSP application significantly decreased the total N_2_O emissions by 49.8% compared to the Control pile (Figure 1a), as compared with the 22.2–55.4% N_2_O mitigation rates in response to CaSSP or biochar addition reported previously [15,17,21,24], the amendment of CaSSP can be recognized as an effective strategy for reducing GHG from composting.

It is known that N_2_O emissions during composting are a microbe-mediated process, which is a balance between production (NO_3_^−^ to N_2_O as catalyzed by nitrite reductase encoding by *nirS*/*nirK*) and consumption (N_2_O to N_2_ as driven by nitrite reductase encoding by *nosZ*) [28]. In this study, CaSSP addition repressed the *nirK* and *nirS* abundance while increased *nosZ* copy numbers, as compared with the Control composting pile, which can explain the N_2_O mitigation based on the microbial functional genes relevant to N-cycle. It has been addressed that biochar, another popular amendment for reducing N_2_O production, also decreased *nirK* population in manure composting [17,24], or *nirS* density under field condition [41,42], and suppressed *nosZ* abundance [17].

The following question is which compost physicochemical properties can mediate the denitrifier community after CaSSP application. Correlation analyses showed that the abundance of *nirK* shared negative relationship with NH_4_^+^-N, moisture, TOC and C/N ratio, and positive correlations with TN as well as NO_3_^−^N. In contrast, the copy number of *nosZ* revealed positive and negative relationship with NH_4_^+^-N and NO_3_^−^-N, respectively (Table 2). Comparison of these characteristics between Control and CaSSP-added piles suggests that NH_4_^+^-N and NO_3_^−^-N (as well as TN) are the major factors that shaped the population of *nirK* and *nosZ*. In addition, the NH_4_^+^-N concentrations of CaSSP treatment were higher than that of CK in the composting process, indicating that CaSSP had a considerable effect on NH_4_^+^-N adsorption and retention. The contents of NO_3_^−^-N maintained a low level from initial stage to the 31th day after composting, but increased at the end of the experiment. This result coincided with Zeng et al. [43] who reported that nitrifying bacteria multiplied with the decrease of temperature and biodegradation rate would promote the transformation of NH_4_^+^-N to NO_3_^−^-N. Additionally, a growing number of studies have illustrated that changes in physicochemical parameters, such as TN, NH_4_^+^-N and NO_3_^−^-N, drove different time course patterns of N_2_O-related functional genes [44,45,46,47], which are consistent with our observation.

Additionally, previous studies had indicated that CaSSP could affect the composting conditions such as reducing pH, which led to an adverse effect on microbial growth [48,49,50]. Results in this study illustrated that the decreased pH brought by CaSSP application did not show significant negative effects on N-cycle functional gene abundance (Table 2). Interestingly, moisture content was found to be negatively correlated with AOA *amoA*, *nirS*, and *nirK* abundance.

Based on physicochemical and biological variables measurements during the composting and their correlation analyses, a schematic model was developed for explaining the dynamics of N_2_O fluxes associated with bacterial functional genes and physicochemical parameters during manure composting (Figure 5). In the schematic model, NH_4_^+^-N and NO_3_^−^-N are correlated with each other and interacting to shape the dynamics of bacterial functional gene abundance. Given this, there is a strong correlation between *nirK* genes and N_2_O fluxes. The present data advised particularly that the abundance of *nirK* genes was likely to play a key role in controlling N_2_O production.

In conclusion, results of this study suggested that calcium superphosphate (CaSSP) addition mediated NH_4_^+^-N/NO_3_^−^-N transformation in pig manure composting, consequently suppressed the abundance of *nirK* genes and significantly decreased N_2_O emissions. Therefore, the use of CaSSP is an effective and economical way to reduce environmental risks and improve compost quality. Additionally, using high-throughput sequencing and a culturomics strategy to determine the functional microbial groups involved in the CaSSP-mediated N_2_O mitigation process, as well as investigating their response to the changing pie physicochemical factors and detailed contribution to N_2_O emissions, will be of great importance for us to deeply understand the GHG mitigation mechanisms mediated by CaSSP, and should be performed in the future.

## Figures and Tables

**Figure 1 ijerph-18-00171-f001:**
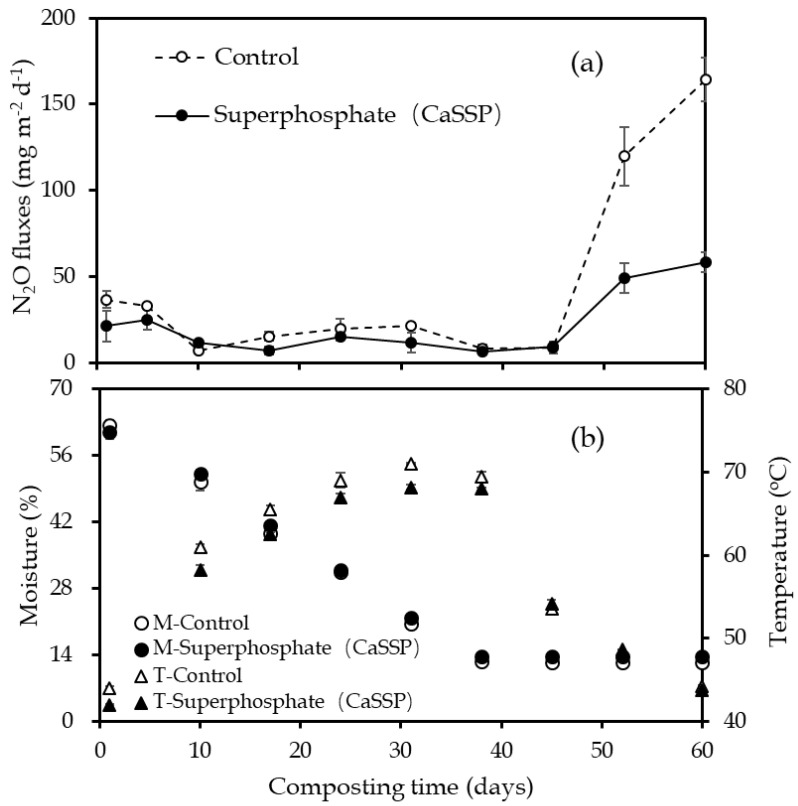
Dynamics of the N_2_O emission rates (**a**), temperature (T) and moisture (M) (**b**) during the pig manure composting experiment. Error bars show standard error of the mean of triplicate compost windrows. In (**b**), “M-Control” and “M-Superphosphate” represent the composting moisture of Control and CaSSP, respectively; “T-Control” and “T-Superphosphate” are the composting temperature of Control and CaSSP, respectively.

**Figure 2 ijerph-18-00171-f002:**
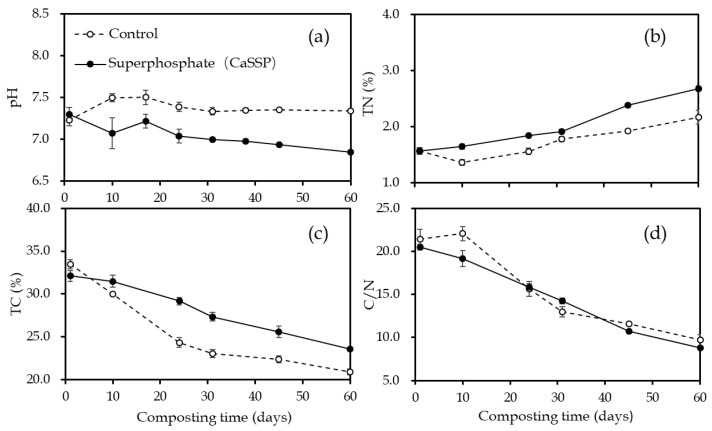
Changes of pH (**a**), total nitrogen (**b**), total carbon (**c**) and C/N (**d**) during composting. Error bars show standard error of the mean of triplicate compost windrows.

**Figure 3 ijerph-18-00171-f003:**
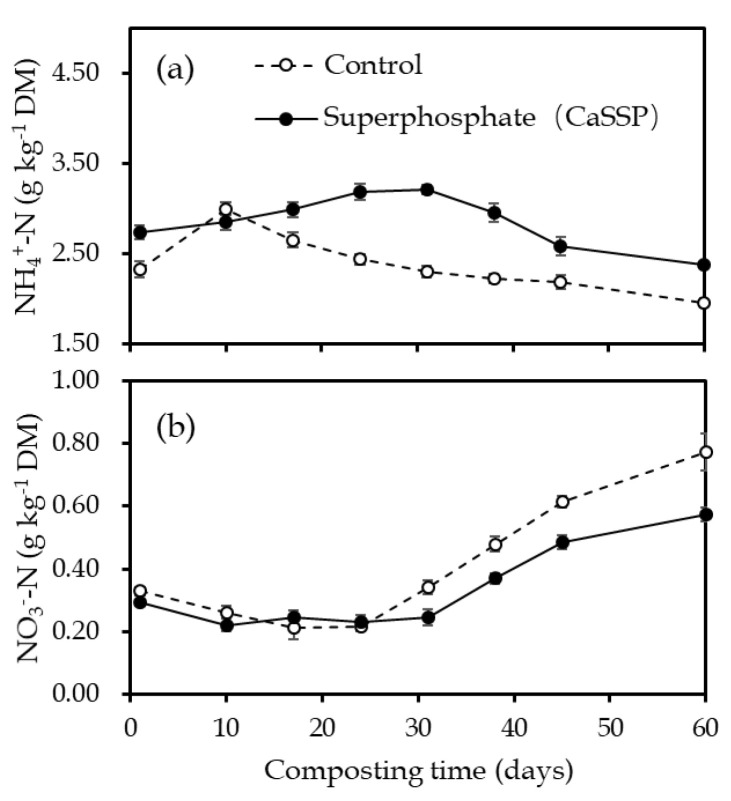
Changes of NH_4_^+^-N and NO_3_^−^-N contents during the pig manure composting experiment. Error bars show standard error of the mean of triplicate compost windrows. (**a**) NH_4_^+^-N content; (**b**) NO_3_^−^-N contents.

**Figure 4 ijerph-18-00171-f004:**
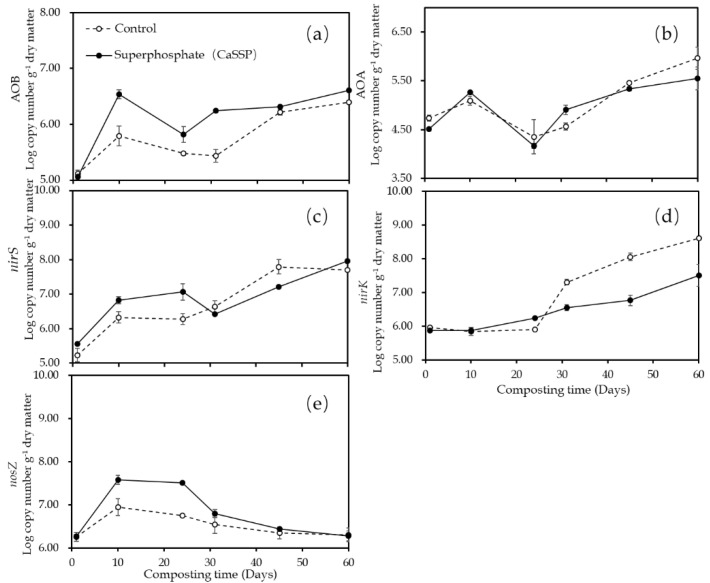
Changes in gene copy numbers per gram of compost (dry matter) for AOA *amoA*, AOB *amoA*, *nirS*, *nirK* and *nosZ* at the days 1, 10, 24, 31, 45 and 60 of composting for Control and CaSSP. Error bars show standard error of the mean of triplicate qPCR reactions. (**a**) AOB *amoA*; (**b**) AOA *amoA*; (**c**) *nirS*; (**d**) *nirK*; (**e**) *nosZ*.

**Figure 5 ijerph-18-00171-f005:**
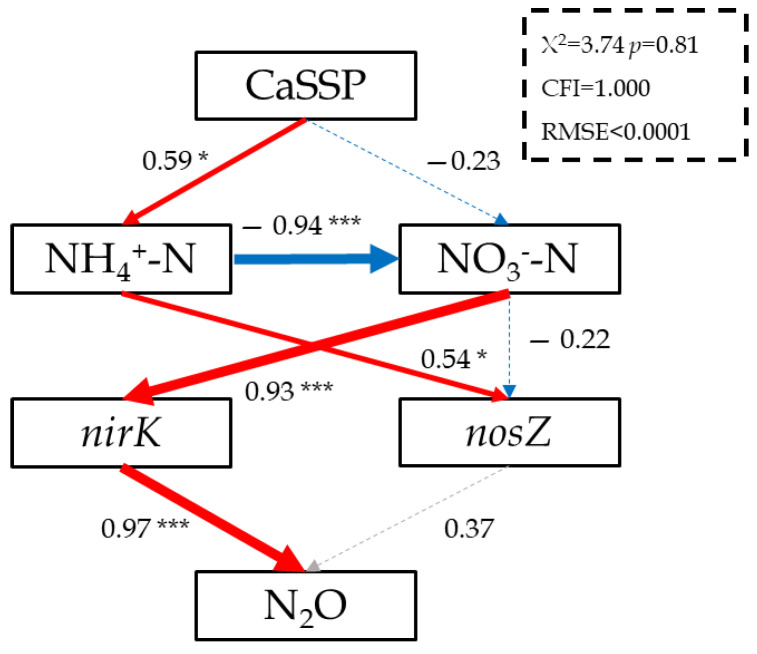
Structural equation model linking physicochemical factors and N-cycle functional genes abundance with N_2_O emissions. Red arrows denote for positive effects while blue arrows denote for negative effects; arrow widths correspond with the relative effect size of each variable. * and *** indicate significance at the 0.05 and 0.001 probability level, respectively.

**Table 1 ijerph-18-00171-t001:** The information of PCR primers used in this study.

Target Gene	Prime Name	Sequence (5′ to 3′)	Product Size	Reference
AOA *amoA*	CrenamoA23f	ATGGTCTGGCTWAGACG	593 bp	[35]
	CrenamoA616r	GCCATCCATCTGTATGTCCA		
AOB *amoA*	*amoA*-1F	GGGGHTTYTACTGGTGGT	491 bp	[36]
	*amoA*-2R	CCCCTCKGSAAAGCCTTCTTC		
*nirS*	*nirS*-Cd3aF	GTSAACGTSAAGGARACSGG	425 bp	[37]
	*nirS*-R3cd	GASTTCGGRTGSGTCTTGA		
*nirK*	*nirK*-F1aCu	ATCATGGTSCTGCCGCG	473 bp	[38]
	*nirK*-R3Cu	GCCTCGATCAGRTTGTGGTT		
*nosZ*	*nosZ*-F	AGAACGACCAGCTGATCGACA	380 bp	[39]
	*nosZ*-R	TCCATGGTGACGCCGTGGTTG		

Note: Y = C/T, S = C/G, V = A/C/G, W = A/T, H = A/C/T, K = G/T, R = A/G.

**Table 2 ijerph-18-00171-t002:** Correlations between the abundance of N_2_O-forming and consuming microbial functional genes and compost physicochemical properties with calcium superphosphate (CaSSP) addition during composting.

	Correlation Coefficients (*r*)								
Parameters	NH_4_^+^-N	NO_3_^−^-N	pH	Moisture	Temperature	Total N	Total C	C/N	N_2_O
AOB	0.00	0.23	−0.49	−0.55	−0.11	0.711 **	−0.41	−0.63	0.35
AOA	−0.45	0.70 *	−0.16	−0.63 *	−0.45	0.644 *	−0.48	−0.60 *	0.42
*nirS*	−0.31	0.57	−0.34	−0.83 **	−0.15	0.86 ***	−0.71 *	−0.82 ***	0.64
*nirK*	−0.66 *	0.81 ***	−0.21	−0.87 ***	−0.14	0.83 ***	−0.83 ***	−0.89 ***	0.79 **
*nosZ*	0.63 *	−0.67 *	0.07	0.11	0.741 **	−0.23	0.08	0.24	−0.42

* Indicates significance at the 0.05 probability level. ** Indicates significance at the 0.01 probability level. *** Indicates significance at the 0.001 probability level (*n* = 3).

## Data Availability

The data presented in this study are available on request from the corresponding author. The data are not publicly available due to conditional confidentiality of the requirements by the National environmental funding project.

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
