# Peer review of "Calcium Superphosphate-Mediated Reshaping of Denitrifying Bacteria Community Contributed to N_2_O Mitigation in Pig Manure Windrow Composting"

_ijerph, 2020, doi:10.3390/ijerph18010171_

Round 1
Reviewer 1 Report
Line 63 gene abundance: state if nitrification is performed by microbes only and if there are known gene involved
Line 88 what kind of straw?
Line 89 DW?
Line 130 It is stated “Nitrification processes were analyzed by…” but here it is about an estimative of Line the presence/activity of the gene, not the overall processes
Line 138, 139 List the primers, or comment the sequences choice.
Line 162 I recommend to initiate the paragraph with “Composting moisture and temperature….” Because it took a time for me to find out that the ‘M’ in the legend refers to moisture.
Line 173 I recommend to write ‘total carbon’ and ‘total nitrogen’ in full, not just ‘TC’ and ‘TN’
Line 181 Obviously or ‘as expected’? Additives? I understood previously that Ca2 (H2PO4)2 was added alone.
Author Response
Response to Reviewer 1
1. Line 63 gene abundance: state if nitrification is performed by microbes only and if there are known gene involved.
Response: Thanks for your question. Nitrification is major performed by microbes and also can be chemocatalysis. The nitrification marker gene amoA is introduced in the revised manuscript.
2. Line 88 what kind of straw?
Response: Sorry for unclearness. It is the mixture of vegetable (Broccoli) and wheat straw, which has been elucidated in the revised manuscript.
3. Line 89 DW?
Response: Sorry for unclearness. “DW” is the abbreviation for “dry weight”, and it has been elucidated in the revised manuscript.
4. Line 130 It is stated “Nitrification processes were analyzed by…” but here it is about an estimative of Line the presence/activity of the gene, not the overall processes.
Response: Thanks for your suggestion. This sentence has been rewritten as “Ammonium monooxygenase gene (amoA) was selected as nitrify marker gene, which include the ammonia oxidizing bacteria (AOB) and archaea (AOA); while the nitric oxide reductase gene (nirK/S) and nitrous oxide reductase gene (nosZ) were targeted as denitrify marker genes.”
5. Line 138, 139 List the primers, or comment the sequences choice.
Response: Sorry for unclearness. The primers and references information have been listed in Table 1 in the revised manuscript.
6. Line 162 I recommend to initiate the paragraph with “Composting moisture and temperature….” Because it took a time for me to find out that the ‘M’ in the legend refers to moisture.
Response: Sorry for unclearness. The meaning of “M” and “T” in Figure 1b has been explained in the legend.
7. Line 173 I recommend to write ‘total carbon’ and ‘total nitrogen’ in full, not just ‘TC’ and ‘TN’
Response: Thanks. It has been revised as suggested.
8. Line 181 Obviously or ‘as expected’? Additives? I understood previously that Ca2 (H2PO4)2 was added alone.
Response: Thanks for your suggestion. It has been revised as “As expected, NH4+-N contents of CaSSP treatment were slightly higher than the Control,…”.

Reviewer 2 Report
Dear authors,
Congratulations on a fine article, but it is slightly amended. A list of comments is given below.
1.Abstract:
Line 18. Please rewrite. Please avoid personal pronouns like 'we'.
2. Introduction:
Please provide statement of novelty. Please describe clearly what is new in your article, compared to the research carried out by other scientists in the quoted articles.
3. Materials and methods:
Line 141-147: Before a post hoc test is carried out, the normal distribution of results and the homogeneity of variances must be checked. Please give the name of the test used to indicate the above mentioned parameters.
4. Results:
Line 175-178: Please explain why TN increased, while emission of N2O occured.
Line 206: Please improve a quality of figure for better readability.
Line 210-211. It seems that the sentence is unfinished.
5. Discussion:
Please redraft the section avoiding personal pronouns and using scientific sentence structures.
Author Response
Response to Reviewer 2
1.Abstract: Line 18. Please rewrite. Please avoid personal pronouns like 'we'.
Response: Thanks for your suggestion. It has been rewritten as “Here a pig manure windrow composting experiment lasting for ~60 days was performed…”.
2. Introduction: Please provide statement of novelty. Please describe clearly what is new in your article, compared to the research carried out by other scientists in the quoted articles.
Response: Thanks for your suggestion. The novelty of this study has been stated as “We successfully identified that CaSSP addition mediated NH4+-N/NO3--N transformation in pig manure composting, consequently suppressed the abundance of nirK genes and significantly decreased N2O emissions; which firstly explain the microbiology mechanism involved in CaSSP mediated N2O mitigation in composting.” in the revised Introduction.
3. Materials and methods: Line 141-147: Before a post hoc test is carried out, the normal distribution of results and the homogeneity of variances must be checked. Please give the name of the test used to indicate the above mentioned parameters.
Response: Sorry for unclearness. The normal distribution of results and the homogeneity of variances have been checked using the Kolmogorov-Smirnov (K-S) and Shapiro-Wilk (S-W) test; this information has been provided in the revised manuscript.
4. Results: Line 175-178: Please explain why TN increased, while emission of N2O occured.
Response: Thanks for your question. During the whole composting process, TN continuously increased because of the less consumption rate as compared with TC. On the other side, major N2O emissions were observed to be occurred after 52 days because of the accumulation of NO3-, as well as the enrichment of nirS/K genes and decline of nosZ gene abundance.
5. Line 206: Please improve a quality of figure for better readability.
Response: Sorry for unclearness. The quality of Figure 4 has been significantly improved in the revised manuscript.
6. Line 210-211. It seems that the sentence is unfinished.
Response: Sorry for the mistake. This should be the title of Chapter 3.4.
7. Discussion: Please redraft the section avoiding personal pronouns and using scientific sentence structures.
Response: Thanks for your suggestion. We have redrafted the Discussion avoiding personal pronouns and using scientific sentence structures.

Reviewer 3 Report
Dear authors,
I have studied your manuscript with great interest because I believe that the subject chosen for the study described here is of great scientific interest and with immediate applicability in practice. I appreciated the experimental design and the presentation of the results, although the article is difficult to follow. I would suggest the introduction of an additional list of abbreviations. There are many in the manuscript and I think such a list would be useful for readers.
Please review the sampling part for q-PCR analyzes: the number of samples, when they were taken, how they were handled prior to DNA extraction. These are very important factors for interpreting the results.
You also state that: moisture content was found to be negatively correlated with AOA, nirS, and 269 nirK abundance, but you have not discussed this at all.
I think it would be interesting to have made a microbiological identification (even based on DNA analysis) to draw clear conclusions about gene abundance. A correlation between the number and type of bacteria found in the samples can support the results obtained, the abundance of genes being closely correlated with the type of bacteria present in the samples. Please better discuss the choices made in this regard.
Author Response
Response to Reviewer 3
1. Please review the sampling part for q-PCR analyzes: the number of samples, when they were taken, how they were handled prior to DNA extraction. These are very important factors for interpreting the results.
Response: Real-time quantitative PCR (qPCR) was performed for investigation of the functional microbial community dynamics during the composting process (days 1, 10, 24, 31, 45, and 60). The composting samples were stored at -80 °C prior to DNA extraction.
2. You also state that: moisture content was found to be negatively correlated with AOA, nirS, and nirK abundance, but you have not discussed this at all.
Response: Thanks for your suggestion. Moisture is an important factor that affects oxygen condition, microbial activities, and N2O emissions. The focus of this study is to discuss the physicochemical and microbial properties changing in response to CaSSP addition; however, no significant differences of moisture content were observed between Control and CaSSP piles, thus this point is not discussed in detail in the manuscript.
3. I think it would be interesting to have made a microbiological identification (even based on DNA analysis) to draw clear conclusions about gene abundance. A correlation between the number and type of bacteria found in the samples can support the results obtained, the abundance of genes being closely correlated with the type of bacteria present in the samples. Please better discuss the choices made in this regard.
Response: Thanks very much for your suggestion. Using high-throughput sequencing and culturomics strategy to determine the functional microbial groups that involved in the CaSSP-mediated N2O mitigation process, as well as investigating their response to the changing pie physicochemical factors and detailed contribution to N2O emissions, will be of great importance for us to deeply understand the GHG mitigation mechanisms mediated by CaSSP, and we are planning to performed these experiments in the future. These choices have been discussed in the revised manuscript.

Reviewer 4 Report
Dear authors,
thank you for your intersting paper. Please include a figure for showing the experimental sampling of gases. Please see my comments in the pdf.
Best regards,
the reviewer

Author Response
Response to Reviewer 4
Please include a figure for showing the experimental sampling of gases. Please see my comments in the pdf.
Response: Thanks very much for your suggestions. We have carefully revised the manuscript following your comments, please see our revised manuscript. Additionally, a Figure for showing the experimental sampling of gases has been included as Figure S1.
Figure S1 The pictures of the experimental static chamber system for gas sampling. (A) The PVC chamber bases (30 cm length × 30 cm width × 25 cm height) were inserted 25 cm into the pile at 10–12 h before gas sampling. (B) The opaque chamber (diameter 0.45 m, height 5 m) was placed on the peak of each windrow with rim of the chamber fitted into the groove of collar. As sampling, the groove in the top edge of the collar was filled with water to seal the rim of the chamber.
Please indicate what the non-sigificance of chi2, CFI and RMSE do mean?
Response: Chi2 and P value are the results of the chi-square test for the structural equation model, and P value stands for the significance of Chi2 value. P>0.05 means the non-significance of chi2 value, and the bigger P value indicated that the structural equation model can better describe the actual data. The CFI is a fit index that measures goodness of fit of the hypothesized model compared to a baseline model. The RMSEA provides a measure of lack of fit in the population with an adjustment for the parsimony of the model.
